# PGCN-DDI: Drug-Drug Interaction Prediction Based on Multidimensional Drug Features and Neighborhood Overlap Similarity

## ABSTRACT

The co-administration of drugs may lead to adverse drug interactions, posing risks to the organism. Therefore, predicting potential drug interactions is crucial. Compared to in vitro experiments and clinical trials for DDI prediction, computational methods are widely used due to their efficiency and other advantages. We propose a deep learning technique, namely the PGCN-DDI model, which enhances the traditional GCN's message passing function through a neighborhood overlap similarity algorithm and improves the aggregator using the Pearson correlation coefficient, ultimately enhancing the model's capability to represent drug node features. We utilize multidimensional features of drugs, including chemical substructure, metabolic pathways, targets, drug ingredients, and drug categories, as model inputs for DDI prediction. We conduct experiments on different feature sets to assess the amount of information contained in different features, leveraging PGCN-DDI to learn drug node features. The results indicate that the combination of drug features (i.e., chemical substructure, targets, and metabolic pathways) outperforms other features in DDI prediction. The experimental results demonstrate that the PGCN-DDI model achieves an accuracy of 0.8921, an AUC of 0.9458, and an AUPR of 0.9208, all of which show improvements over several baseline models.

## CCS CONCEPTS

• **Applied computing** → *Consumer health.*

## KEYWORDS

drug–drug interaction prediction, multidimensional features, neighborhood overlap similarity

## 1 INTRODUCTION

Although combining medications is common in treating complex diseases or processes requiring synergistic therapy, the possibility of adverse effects from interacting drugs, known as Drug-Drug Interactions (DDIs), poses a risk to patient lives. DDIs, often caused by chemical-physical interactions between co-administered drugs, contribute significantly to drug withdrawals and incur substantial healthcare costs. Therefore, DDIs have become a focal point in clinical research. This study falls into the category of DDI prediction, aiming to forecast interactions between pairs of drugs. Automated computational methods, such as machine and deep learning, are crucial in addressing this issue due to limitations in manually screening DDI candidates during clinical trials and handling the rapid growth of biomedical data. This work proposes a deep learning-based automated computational approach for predicting DDIs between pairs of drugs. Leveraging graph convolutional neural networks with enhanced message-passing functions and aggregators, it learns graph representations of drugs' multidimensional features. In the message-passing step, drug node features are scaled based on the similarity of neighborhood overlaps. A neighborhood overlap similarity algorithm is proposed to derive weighting coefficients for scaling. In the aggregation stage, the linear correlation between feature vectors of two drug nodes in the feature space is considered. Pearson correlation coefficient measures the linear correlation between feature vectors of two drug nodes, allowing nodes with stronger linear correlations to have higher contributions.

## 2 RELATED WORK

Methods based on automated computation can roughly be classified into the following categories: (1) methods based on Natural Language Processing (NLP), (2) methods based on matrix factorization, (3) methods based on machine learning,(4) methods based on deep learning.

NLP-based methods utilize a large amount of text data to mine and analyze drug-drug interaction (DDI) information using domain knowledge, clinical evidence, and automated techniques. They can quickly retrieve, filter, and integrate scattered information, discover patterns and trends hidden in a large number of literature, and improve computational models with more external knowledge to enhance predictive performance and interpretability. Asada et al. proposed a new method for DDI extraction that effectively utilizes external drug database information and information from large-scale pure text. They focus on drug descriptions and molecular structure information as drug database information[1]. Liu et al. proposed a machine learning framework to extract useful features from FDA adverse event reports and then used a semi-supervised learning algorithm based on autoencoders to identify potential high-priority DDIs[2].Zhao et al. proposed a DDI extraction method based on Syntax Convolutional Neural Networks (CNNs) and achieved better performance than other state-of-the-art methods[3].

Matrix factorization methods include non-negative matrix factorization, singular value decomposition, principal component analysis, LU decomposition, etc., and the DDI prediction task is akin to matrix completion tasks. Zhang et al. proposed a manifold regularized matrix factorization method for DDI prediction[4]. Yu et al. developed a novel method called DDINMF based on semi-non-negative matrix factorization[5]. Zhu et al. designed a dependency network to model drug dependencies and proposed a probabilistic matrix tri-factorization model for DDI prediction[6].

Research methods based on machine learning for predicting DDI are very diverse. For example, Kstrin et al. considered both topological and semantic feature similarities and used five classifiers to predict drug-drug interactions[7]. Qian et al. constructed a gradient boosting classifier using feature similarity and feature selection

methods to speed up the process and achieve robust predictive performance[8]. Gottlieb et al. computed seven types of similarities and combined the two best similarities for each drug pair to generate a feature[9]. Cami et al. used standard ensemble methods to combine multiple predictors for DDI prediction[10]. The authors constructed a DDI network and obtained multiple covariates from the network to build logistic regression models and generalized linear mixed models. Chen et al. extracted features from simplified molecular input line entry system data and drug pair adverse effect similarity and applied support vector machines (SVMs) to predict DDI[11].

Models based on deep learning mainly include models based on graph embeddings, models based on deep neural networks (DNNs), and models based on knowledge graph embeddings. Compared to traditional machine learning and matrix factorization, deep learning can automatically extract features, reducing human intervention. It excels in handling complex data features and large-scale datasets, resulting in higher model accuracy. Rohani et al. computed various drug similarities and Gaussian interaction curves for drug pairs, applied this method to select features with maximum information and less redundancy, then used the feature vectors of drug pairs as inputs to neural networks for prediction[12]. Zitnick et al. proposed a method to predict drug side effects, considering DDI prediction as a multi-modal graph problem involving multiple relationship chains on the graph, including relationships between drugs, proteins, and side effects[13]. They used Graph Convolutional Networks (GCN) as encoders to generate embeddings of nodes on the graph and used tensor factorization models as decoders to predict DDIs. Additionally, this work extends GCN to graphs with multiple node types and multiple edge types. As part of deep learning, Graph Neural Networks (GNNs) are particularly adept at processing graph-structured data. GNNs have many applications, for instance, Gupta et al. proposed a GNN model that integrates spatial, topological, and temporal information into node representations to model spatiotemporal processes in road networks, addressing various challenges[14]. Ding et al. proposed a fine-grained IP geolocation framework based on GNNs[15]. Wang et al. used GNNs to predict the impact of mutations on protein stability[16]. The ability of GNNs to extract features from graph-structured data is widely recognized by many researchers. Using GNNs as a tool for drug interaction prediction allows for the comprehensive learning of the topological features of drug interaction networks and the intrinsic features of drug nodes. This paper proposes an algorithm that enhances the message passing and message aggregation functions of GNNs to improve their learning ability, thereby increasing the accuracy of drug interaction prediction.

## 3 PROPOSED WORK

GCN is a subset of GNN. GCN aggregates information from nodes and their neighborhoods by introducing convolution operations on graph-structured data, generating node representations. We take GCN as tool for our study. GCN progressively updates node feature representations through multiple rounds of message passing and aggregation, utilizing information from nodes and their neighbors to learn each node's representation. The process comprises three main parts: message passing, message aggregation, and node representation updating. This paper mainly contributes to the first two parts, illustrated in Fig. 1. In summary, this method aims to reveal the importance of nodes in a different way by considering the linear correlation between nodes relative to the target node and the degree of overlap similarity in their neighborhoods.

### 3.1 MessagePassing Function

In traditional GCN, each layer defines a matrix that specifies how node features are transformed before being passed to the next layer. In other words, the message passing function simply forwards node features to the next layer. In contrast, the model proposed in this paper scales node features during the message passing step based on the degree of neighborhood overlap similarity. To achieve this, a neighborhood overlap similarity algorithm is introduced to obtain the weight coefficients used for scaling.The algorithm is as follows:

Firstly, calculate the neighborhood overlap similarity between the central node and its first-order neighboring nodes using the following formula:

$$J(a,b) = \frac{|X \cap Y|}{|X \cup Y|} \times \alpha$$

Where $X$ is the first-order neighborhood set of the central node $a$ in graph $G$, and $Y$ is the first-order neighborhood set of the neighbor node $b$ in graph $G$. Ultimately, an overlapping neighborhood similarity matrix is obtained, denoted by $Gather\_sims$:

$$Gather\_sims = \{J_1, J_2, J_3, \ldots, J_{|E|}\}$$

with dimensions $|E|x1$.The matrix $Gather-sims$ is processed by inputting it into a weight function inversely proportional to the neighborhood overlap similarity. This weight function is defined as:

$$GJ(a,b) = \left( \frac{1+e^{-J(a,b)}}{2}, \frac{1+e^{-2J(a,b)}}{2}, \ldots, \frac{1+e^{-NJ(a,b)}}{2} \right)^T$$

The term $\frac{1+e^{-J(a,b)}}{2}$ is used to map the original values to a probability space, where the range is mapped from $[0,1]$ to $[0.5,1]$. This adjustment increases the confidence in the occurrence of an event, a process known as calibration in statistics. In probability theory, probabilities within the range $[0.5,1]$ are generally considered closer to the true probabilities of events, as they correspond to higher confidence levels. This adjustment imbues the resulting vector with a more dynamic and rich characteristic, allowing for better expression of similarities and differences between data. It is commonly used to calibrate model outputs to better align with real-world scenarios and improve predictive performance.$N$ is a positive integer. The obtained vector $G_J(a,b)$ is of size $|E|xN$, which is then fed into a perceptron for nonlinear transformation, merging, and integration of various features and details from the original high-dimensional vectors:

$$fg(a,b) = \tau(W_{MLP} \cdot GJ(a,b) + bias)$$

Here, $\tau(\cdot)$ represents the activation function, and $bias$ is a constant parameter.Finally, $fg(a,b)$ is input into a dropout layer to output the final weight coefficients $F(a,b)$.The weight coefficients

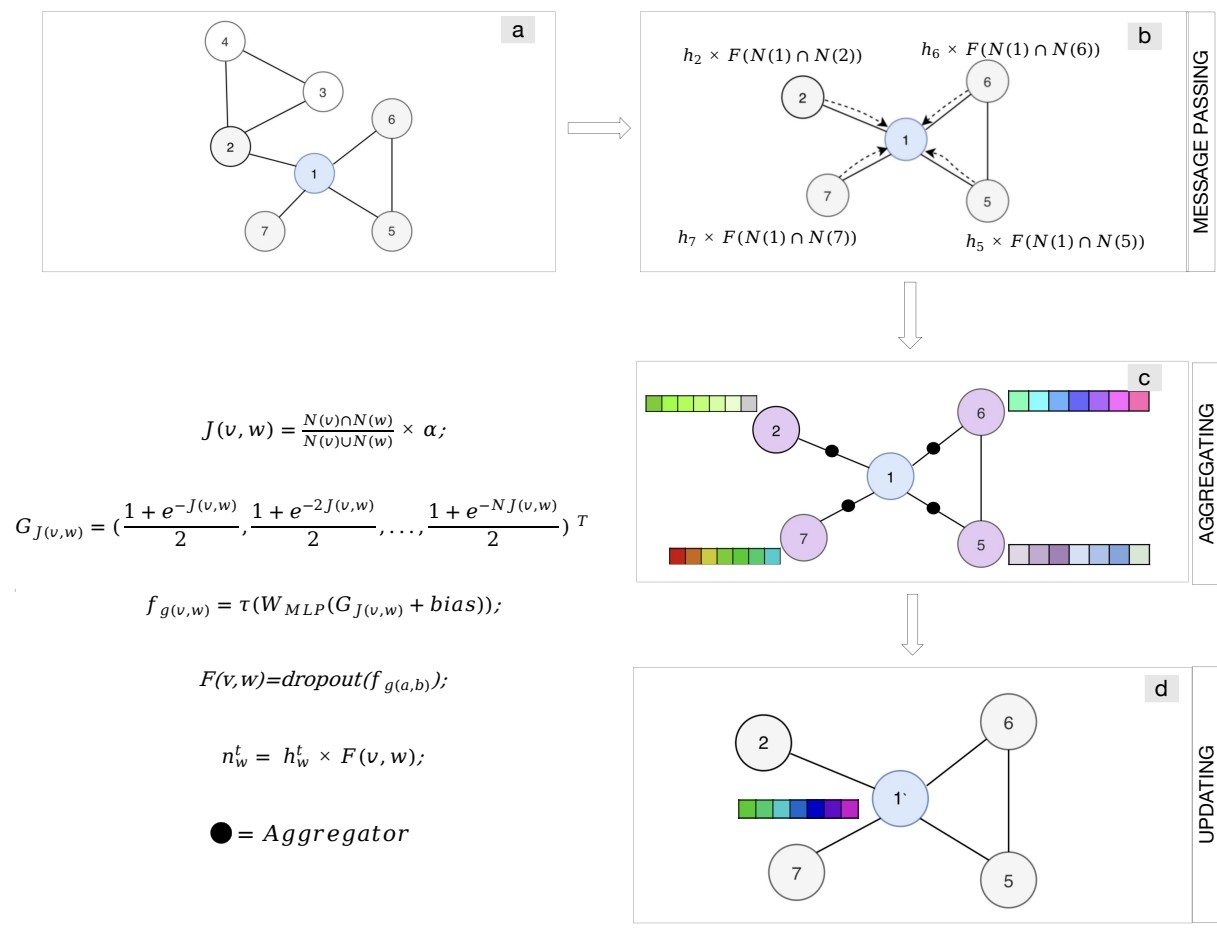

**Figure 1: Message passing and message aggregation process**

are then incorporated into the message passing process and scaled using the weight coefficients. The scaling formula is as follows:

$$n_{wt} = F(w,v) * h_{wt}, \quad w \in N(v)$$

$$m_{vt} = (n_{wt}, e_{vw})$$

## 3.2 Aggregation Function

After message passing, the central node's mailbox receives information from k-hop neighbors, which is aggregated using a function. Traditional aggregation functions in graph GCNs are simple mean and pooling. Specifically, mean aggregation computes the average of neighbor features, while pooling aggregation performs pooling operations (e.g., max or mean pooling) on neighbor features to obtain a summarized representation. In this work, an improvement is proposed for the GCN model's aggregation function. Before aggregation, weights are assigned to each node, calculated using the

Pearson correlation coefficient, a statistical measure of linear correlation between two vectors ranging from -1 to 1. This coefficient is used to scale neighbor features, considering their linear relationship in feature space. The formula for the Pearson correlation coefficient is:

$$\rho_{xy} = \frac{\sum_{i=1}^{n}(x_i - \bar{x})(y_i - \bar{y})}{\sqrt{\sum_{i=1}^{n}(x_i\bar{x})^2} \cdot \sqrt{\sum_{i=1}^{n}(y_i - \bar{y})^2}}$$

The magnitude of the Pearson correlation coefficient between two vectors X and Y can be visually presented. For example, when the coefficient is 0.990, the visualization of X and Y is as shown in Figure 2, and when it is 0.894, the visualization is as shown in Figure 3.

$$P = \text{Pearson}(n_w^t, h_v^t)$$

$$m_v^{t+1} = (P * n_w^t, e_{vw})$$

$$h_v^{t+1} = U_t(h_v^t, m_v^{t+1})$$

After scaling neighbor features using the Pearson correlation coefficient as weights, $m_v^{t+1}$ is obtained. $U_t$ is the update function used to update the representation vector $h_v^t$ of node $v$, resulting in the final central node representation $h_v^{t+1}$. In Figure 1, there is an example where the central node is numbered 1, and it has four neighboring nodes, numbered 2, 5, 6, and 7, as shown in step a. Step b is the message passing process, where neighboring nodes transmit their feature information to node 1. Before transmission, a weight coefficient is obtained based on the neighborhood overlap similarity algorithm. This weight coefficient scales the feature information, which is then transmitted to the mailbox. In step c, the aggregator calculates the Pearson correlation coefficient between the scaled feature information and the feature information of node 1, which measures the strength of the linear correlation. This results in another scaling. Finally, in step d, the feature vector of node 1, which aggregates the neighbor information, is updated.

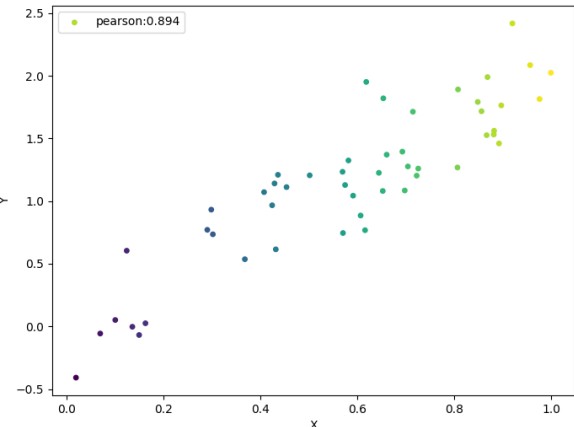

**Figure 3: When the Pearson coefficient equals 0.894.**

into 920 binary digits. As illustrated in Figure 4, each digit of this binary number represents a chemical structure, with a value of 1 indicating the presence of that structure in the drug molecule and 0 indicating absence. We calculated the Tanimoto coefficient between pairs of drugs using these binary numbers. The formula for calculating the Tanimoto coefficient is as follow:

$$T = \frac{|A \cap B|}{|A \cup B|}$$

In this formula, $T$ represents the Tanimoto coefficient, and $A$ and $B$ represent two binary numbers. $|A \cap B|$ represents the number of positions where both sets $A$ and $B$ have a value of 1, and $|A \cup B|$ represents the total number of positions where either set $A$ or $B$ has a value of 1. This computation yields a matrix of drug two-dimensional structural similarity. We then applied Principal Component Analysis (PCA) to reduce the dimensionality of this matrix, resulting in a molecular structure feature vector for each drug. Next, the drug targets, drug categories, drug ingredients, and metabolic pathways are encoded as follow: Iterate through all drugs to determine the common N drug targets. Then, for each drug, perform the following process: For drugs with a certain target, change the corresponding position from 0 to 1 in an N-dimensional zero vector, resulting in an N-dimensional target feature vector for the drug. The same process is applied to drug categories, drug ingredients, and metabolic pathways. Finally, this yields target feature vectors, drug category vectors, drug ingredient vectors, and metabolic pathway vectors. These vectors are concatenated with the molecular structure feature vectors. Principal Component Analysis (PCA) is then applied for dimensionality reduction to obtain the final node feature vectors for each drug.

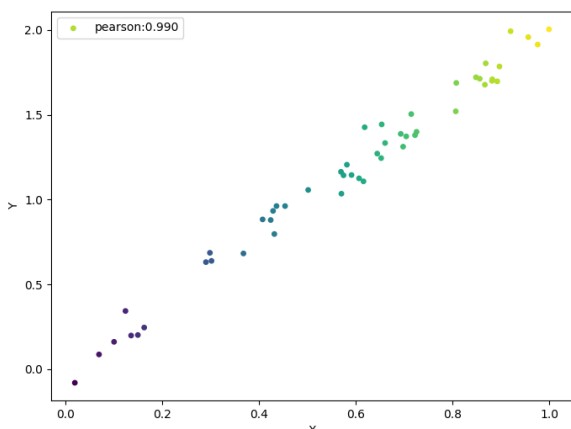

**Figure 2: When the Pearson coefficient equals 0.990.**

The proposed modification is compatible with most existing aggregation functions, such as mean and pooling. However, mean aggregation function is empirically chosen and employed in the current model. Additionally, the aggregation function design adheres to the theory of permutation invariance (Defferrard et al., 2016), meaning it does not depend on the ordering of nodes in the graph. This ensures its compatibility with various aggregation functions to accommodate different graph representations and scenarios.

### 3.3 Extract drug features

In this study, the datasets were sourced from three databases: Drug-Bank, PubChem, and DDinter. We obtained the network topology graph of drug interactions from DDinter, the chemical structure features of drug molecules from PubChem, and the features such as drug targets, drug categories, drug ingredients, and metabolic pathways from DrugBank.

The molecular fingerprints of drugs obtained from the PubChem database consist of 230 hexadecimal digits, which were converted

## 4 EXPERIMENT

This section primarily evaluates the performance of PGCN-DDI on various feature sets and compares it with the state-of-the-art prediction model:DDIMDL. Additionally, we considered several

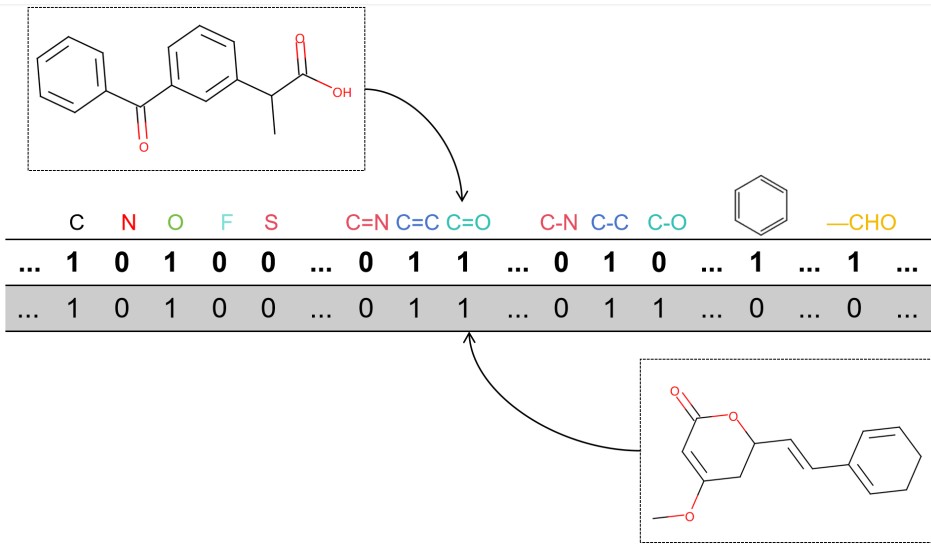

**Figure 4: A schematic diagram of chemical substructure molecular fingerprints.**

popular classification methods, such as DNN and KNN. To demonstrate our advantages, we compared PGCN-DDI with these models.We use ACC, AUC, AUPR, and F1 as model evaluation metrics. ACC (Accuracy) represents the proportion of correctly classified samples out of the total samples. AUC (Area Under Curve) measures the area under the ROC curve, evaluating the classifier's ability to distinguish between positive and negative samples. AUPR (Area Under Precision-Recall Curve) reflects the classifier's performance on imbalanced data by measuring the area under the precision-recall curve. F1 score is the harmonic mean of precision and recall, providing a comprehensive evaluation of the classification performance. In terms of model implementation, there are several open-source frameworks to choose from, most of which are built using the Python programming language, including Deep Graph Library (DGL), PyTorch Geometric, MXNets, Graph Nets, and DIG. We choose to implement and experiment using PyG. All experiments were conducted on a computer equipped with an Intel Core i7-10700K@3.8 GHz processor and an NVIDIA GeForce GTX 3060 Ti GPU. All models were implemented in a Python 3.7.11 environment, trained and tested using the PyTorch framework on CUDA 11.1. To introduce non-linearity and compute losses, we adopted the Softmax function and the cross-entropy loss function. In the previous formula $J(a, b)$, $\alpha$ is an adjustable parameter used to weigh the calculation of Jaccard similarity. By adjusting $\alpha$, the weighting of neighboring nodes in the Jaccard similarity calculation can be controlled. A larger $\alpha$ value emphasizes the importance of nodes that commonly appear among the neighbors of nodes $a$ and $b$, while a smaller $\alpha$ value treats all neighboring nodes more equally. Therefore, the choice of $\alpha$ affects the result of the Jaccard similarity calculation, thereby impacting the calculation of the similarity degree between nodes $a$ and $b$. This $\alpha$ is considered a hyperparameter. We started with an $\alpha$ value of 0.5 and increased it by 0.5 in each experiment. Ultimately, we found that an $\alpha$ value of 1.5 yielded better results than 1.0 and 2.0.

## 4.1 Performance on different feature sets

To evaluate the influence of different feature sets on model performance, we conducted a series of experiments, including using only single drug features and different combinations of drug features. Table 1 shows the results of single features and feature combinations, where S denotes chemical substructures, T denotes targets, C denotes categories, I denotes ingredients, and M denotes metabolic pathways. The accuracy when using only the network topology structure is 0.7611. When chemical substructure features are added, the accuracy increases to 0.8671. The accuracy for metabolic pathways is 0.8217, while for target models it is 0.8321. When only drug ingredients, drug categories, and other features are added, the accuracy ranges from 0.770 to 0.780, indicating relatively less information provided by these features. We conducted experiments on feature fusion based on the top three features ranked by the amount of information provided by single drug features. The accuracy for the fusion of chemical substructure and metabolic pathways is 0.8783, surpassing all other drug features used in other feature combinations. When fusing these three sets of features, the accuracy for chemical substructure, target, and metabolic pathways is 0.8921. Training the model on all combinations of four or more features yields an accuracy of 0.8853. Experimental results indicate that the performance level of drug combinations using all features cannot reach that of the aforementioned three features. In Figure 5, it is evident that when fusing these three sets of features, the remaining metrics for chemical substructure, target, and metabolic pathways are all highest.

## 4.2 Comparing with other methods

In the following experiments, we will train the model using a dataset containing both the network topology information of drug nodes and the feature information of drug nodes, utilizing only three features: chemical substructures, targets, and enzymes, which have

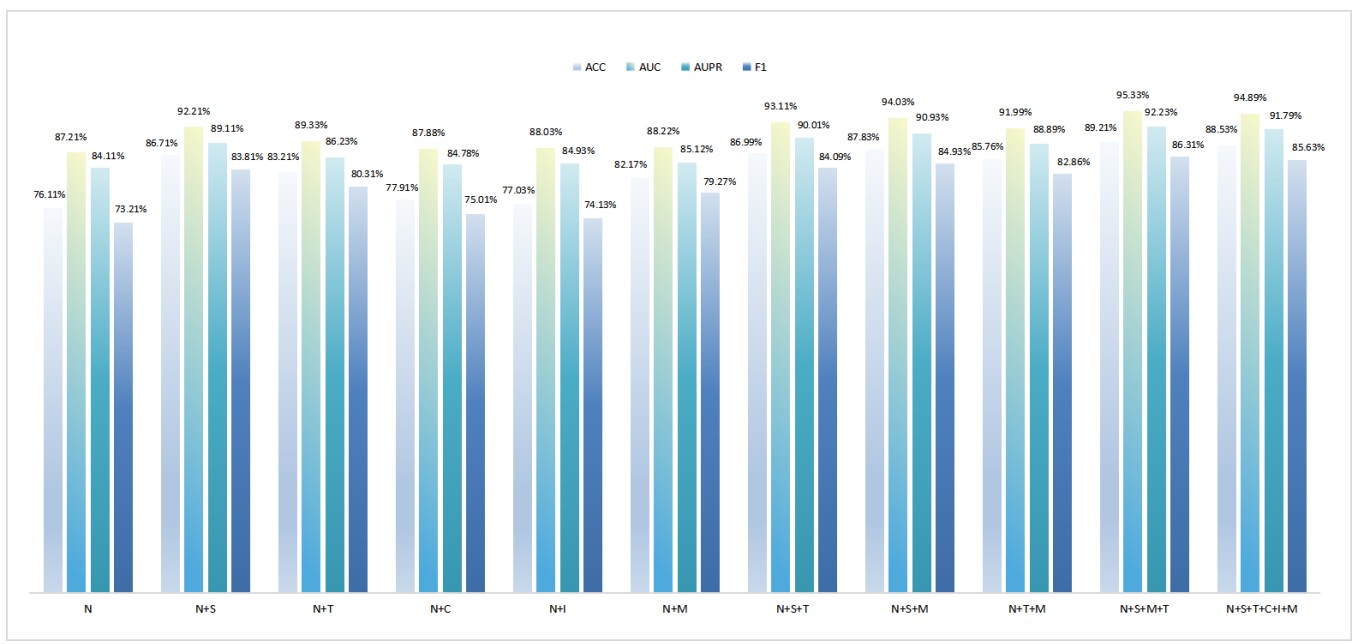

**Figure 5: The four evaluation metrics on different feature sets**

| Fea | ACC | AUC | AUPR | F1 |
|---|---|---|---|---|
| N | 0.7611 | 0.8721 | 0.8411 | 0.7321 |
| N+S | 0.8671 | 0.9221 | 0.8911 | 0.8381 |
| N+T | 0.8321 | 0.8933 | 0.8623 | 0.8031 |
| N+C | 0.7791 | 0.8788 | 0.8478 | 0.7501 |
| N+I | 0.7703 | 0.8803 | 0.8493 | 0.7413 |
| N+M | 0.8217 | 0.8822 | 0.8512 | 0.7927 |
| N+S+T | 0.8699 | 0.9311 | 0.9001 | 0.8409 |
| N+S+M | 0.8783 | 0.9403 | 0.9093 | 0.8493 |
| N+T+M | 0.8576 | 0.9199 | 0.8889 | 0.8286 |
| N+S+M+T | 0.8921 | 0.9533 | 0.9223 | 0.8631 |
| N+S+T+C+I+M | 0.8853 | 0.9489 | 0.9179 | 0.8563 |

**Table 1: Results of the model on different feature sets.**

| Method | ACC | AUC | AUPR | F1 |
|---|---|---|---|---|
| PGCN-DDI | 0.8921 | 0.9458 | 0.9208 | 0.7671 |
| DDIMDL | 0.8733 | 0.9046 | 0.8796 | 0.7483 |
| KNN | 0.7414 | 0.8901 | 0.7922 | 0.6122 |
| DNN | 0.8521 | 0.9023 | 0.8773 | 0.7271 |

**Table 2: Comparison results with other methods.**

demonstrated better performance. This experiment will be compared with methods such as DDIMDL, KNN, and DNN, where the neighbor value for KNN is set to 3. In Figure 6, it can be observed that PGCN-DDI encloses a larger area with the coordinate axes compared to the other methods. The experimental results of PGCN-DDI and the other models are presented in Table 2, where our model achieves high accuracy and AUC values of 0.8921 and 0.9458, respectively. Additionally, both AUPR and F1 are higher than the other three methods, as depicted more intuitively in Figure 7, highlighting the superiority of PGCN-DDI.

## 5 CONCLUSION

In recent years, deep learning techniques have been applied to predict Drug-Drug Interactions (DDI). However, most of these studies have focused on a single feature of drugs or considered only

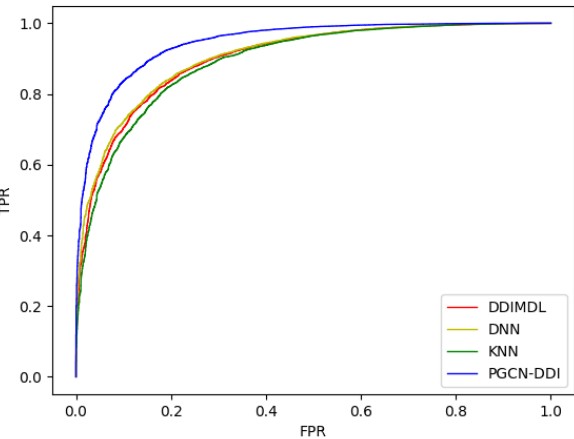

**Figure 6: ROC curve plot comparing results with other methods.**

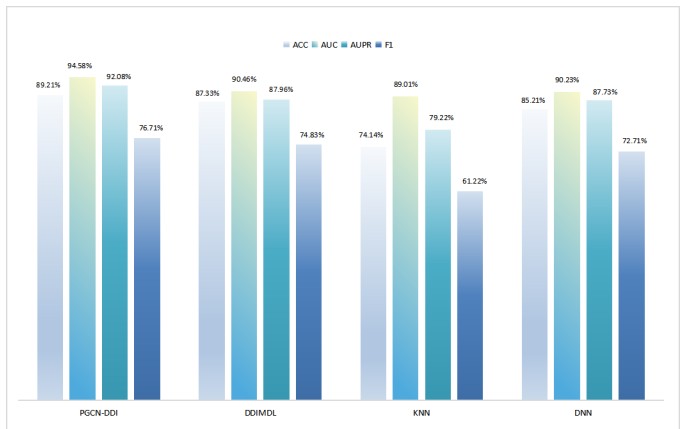

**Figure 7: The four metrics compared with the other methods**

whether one drug interacts with another. In this study, we acquire multidimensional features of drugs from multiple drug databases, as well as the topological features of known drug interaction networks. We propose a neighborhood overlap similarity algorithm to enhance the message passing function and utilize Pearson correlation coefficient to study the linear correlation of drug feature vectors, thereby enhancing the aggregator. Finally, we combine diverse drug features for DDI prediction and demonstrate through experiments that our model achieves better prediction accuracy.

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
