# OpenReview forum: "PGCN-DDI: Drug-Drug Interaction Prediction Based on Multidimensional Drug Features and Neighborhood Overlap Similarity"
_KDD.org/2024/Workshop/AIDSH — KDD-AIDSH 2024 Poster_

### Official Review · Reviewer_mJ6T · 2024-06-16
**Well-written, but lack of novelty in the GCN part, and missing a strong baseline**

**Rating:** 5
**Confidence:** 4

**Review:**

Summary Of Strengths:
1. Simple method for Drug-Drug Interaction Prediction
2. Extensive feature extracting and clear ablation of the importance of these features for the task
3. The paper is well written and easy to follow

Summary Of Weaknesses:
- Lack of comparison to strong baselines, especially GCN-based DDI prediction methods.
- The authors don't provide the ablation studies for using the Pearson correlation coefficient in the GCN framework.

---

### Official Review · Reviewer_fZYH · 2024-06-19
**Reviews for PGCN-DDI, an improved GCN for Drug-Drug Interaction Prediction**

**Rating:** 6
**Confidence:** 4

**Review:**

The research on drug-drug interactions(DDI ) is the basis for avoiding unreasonable drugs in the treatment of complex diseases and is also a hot research point. This work proposes a new graph convolution model PGCN-DDI. The model learns graph representations of drugs' multidimensional features, including chemical substructure, metabolic pathways, targets, drug ingredients, and drug categories. In the message-passing module of the proposed method, a neighborhood overlap similarity algorithm is proposed to derive weighting coefficients for scaling. In the aggregation module, the Pearson correlation coefficient is used as the linear correlation between representations of two drugs. The paper is written clearly. However, it still has the following shortcomings:
1. The innovation of the proposed method is not significant enough. There are many variants of the two-stage improvement of GCN. Whether this method applies to all GCN scenarios or is an improvement specially designed for drug DDI prediction tasks. The improvement principle is not clearly expressed.
2. No citations were found for the state-of-the-art prediction model DDIMDL. The other comparison methods are KNN and DNN. There is a lack of experimental comparison of other DDI prediction methods. The experimental content is relatively weak, including the lack of ablation studies on different improvements and sensitivity analysis of different hyperparameters.

---

### Decision · Program_Chairs · 2024-06-28

Accept (Poster)